# Opportunities and Challenges in Targeting the Proofreading Activity of SARS-CoV-2 Polymerase Complex

**DOI:** 10.3390/molecules27092918

**Published:** 2022-05-03

**Authors:** Jerome Deval, Zachary A. Gurard-Levin

**Affiliations:** 1Aligos Therapeutics, Inc., San Francisco, CA 94080, USA; 2SAMDI Tech, Inc., Chicago, IL 60616, USA; zgurardlevin@samditech.com

**Keywords:** coronavirus, SARS-CoV-2, polymerase, proofreading, NSP14, exonuclease, ExoN, inhibitor, nucleoside

## Abstract

Severe acute respiratory syndrome coronavirus 2 (SARS-CoV-2) is the cause of the COVID-19 pandemic. While the development of vaccines and the emergence of antiviral therapeutics is promising, alternative strategies to combat COVID-19 (and potential future pandemics) remain an unmet need. Coronaviruses feature a unique mechanism that may present opportunities for therapeutic intervention: the RNA polymerase complex of coronaviruses is distinct in its ability to proofread and remove mismatched nucleotides during genome replication and transcription. The proofreading activity has been linked to the exonuclease (ExoN) activity of non-structural protein 14 (NSP14). Here, we review the role of NSP14, and other NSPs, in SARS-CoV-2 replication and describe the assays that have been developed to assess the ExoN function. We also review the nucleoside analogs and non-nucleoside inhibitors known to interfere with the proofreading activity of NSP14. Although not yet validated, the potential use of non-nucleoside proofreading inhibitors in combination with chain-terminating nucleosides may be a promising avenue for the development of anti-CoV agents.

## 1. Introduction

The respiratory infection COVID-19 is caused by severe acute respiratory syndrome coronavirus 2 (SARS-CoV-2). SARS-CoV-2 is an enveloped, positive-sense, single-stranded RNA virus that belongs to the β-coronavirus genus of the Coronaviridae family as part of the Nidovirales order [1]. The RNA genome of SARS-CoV-2 is divided into multiple open-reading frames, among which ORF1ab contains the nonstructural proteins (NSPs) 7 to 16 required for RNA replication and transcription, with the exception of NSP11. The replication and transcription of CoV RNA requires a complex and multi-step coordination of viral enzymatic functions encoded by NSPs (for a complete review, see [2]). NSP12 contains the RNA-dependent-RNA polymerase (RdRp) activity responsible for RNA synthesis and is considered a major target for antiviral agents [3,4]. NSP12 also encodes a second enzymatic site referred to as the NiRAN (nidovirus RdRp-associated nucleotidyltransferase) also responsible for UMPylation of the NSP9 [5]. The NiRAN domain of NSP12 is found only in viruses belonging to the Nidovirales order and is an emerging target for antiviral therapy development [6]. NSP13 encodes nucleoside triphosphatase and helicase functions responsible for unwinding viral RNA. NSP15 encodes an endonuclease activity, and NSP16 is a 2′O-MTase; both enzymes are key to viral RNA maturation. NSP14 is a bifunctional enzyme featuring exonuclease (ExoN) and guanine G7-methylransferase (MTase) activities that plays a critical role in SARS-CoV-2 replication. While the G7-MTase role in RNA capping and its relevance as a therapeutic target have been described elsewhere [7,8,9,10,11], this review focuses on the function and potential therapeutic opportunity of the ExoN proofreading activity of NSP14.

## 2. NSP14 Is Essential to RNA Replication and Transcription

First identified in SARS-CoV, the ExoN activity of double-stranded RNA (dsRNA) proceeds in a 3′ to 5′ direction [12]. Conserved amongst cellular exonucleases [13], the ExoN domain features the critical Asp and Glu residues along with the two metal binding sites that drive nucleotide excision, classifying NSP14 in the DEDDh superfamily [14,15], which includes DNA proofreading enzymes. The identification of NSP14 catalytic residues is further supported by structural studies [16] that also highlight the importance of two zinc finger domains for ExoN activity [17]. A yeast two-hybrid screen revealed that NSP14 interacts with NSP10 [18], an interaction that increased ExoN activity over 35-fold, potentially through stabilization of the NSP14 ExoN active site [16,19].

The combination of biochemical and structural approaches has shed light on the role of NSP14 in viral replication. Pull-down studies using recombinantly expressed proteins in *Escherichia coli* confirmed that the NSP12-NSP7-NSP8 complex can interact with NSP14 with no loss of polymerase, ExoN, or G7-MTase activities [20]. The RdRp protein (NSP12) interacts with the NSP14 ExoN and G7-MTase domains. Small-angle X-ray scattering studies revealed a hinge region that separates the ExoN and G7-MTase domains, suggesting a molecular switch allowing different functions and activities with diverse substrates [21] (reviewed in [22]). These data support a model in which RNA polymerization and proofreading occur in a single complex with a high degree of cooperativity so that nascent double stranded RNA exiting the polymerase passes over the NSP14 ExoN and NSP15 endonuclease sites before being unwound [23]. This process is further supported by recent crystal structures highlighting the interaction between NSP14 and NSP10 along with cryo-EM structures showing how NSP14-NSP10 facilitates proofreading of viral RNA in concert with NSP12 [24,25,26].

The primary role of NSP14 ExoN in the replication complex is to ensure replication fidelity by removing mismatched nucleotides from the 3′ end of the growing RNA strand [21,27]. In mouse hepatitis virus (MHV)-CoV, catalytically inactive (yet viable) ExoN mutants generate significantly altered patterns of recombination and less frequent recombination events than the wild type. Decreased subgenomic RNA populations and increased defective viral genomes observed with these mutants support the importance of proofreading by NSP14 ExoN in viral RNA synthesis and viral fitness [28]. Genetic studies further support the proofreading mechanism of NSP14. Loss-of-function ExoN mutants exhibit a 20-fold increase in mutation frequency during replication in cell culture [13,29].

In addition to its role in proofreading, the NSP14 ExoN activity has also been implicated in the innate immune response. By digesting double-stranded RNA (dsRNA)—an intermediate of viral replication that often triggers an immune response—NSP14 ExoN has been proposed to shield the RNA from recognition by innate immune sensors [30]. This mechanism, together with the function of the G7-MTase domain in mRNA cap formation to mask viral RNA as “self”, contributes to further evade the host immune response [31]. NSP14 is highly conserved in α, β, and γ coronaviruses, which explains why these mechanisms of immune response evasion are also shared by other coronaviruses [32,33,34,35,36]. Taken together, these data support the potential of NSP14 as a therapeutic target for addressing current and future coronavirus pandemics.

## 3. In Vitro Assays to Identify and Characterize NSP14 Proofreading Inhibitors

To date, drug discovery efforts focused on NSP14 have been limited. While there are many established assay technologies well suited for measuring the methyltransferase activity of NSP14, assays that measure the ExoN and proofreading activity are more challenging. Historically, nuclease activities are mainly characterized by gel electrophoresis, which is useful for characterizing enzyme activities and simultaneously providing information on the various enzymatic products, but the limited throughput is not conducive for large-scale drug discovery efforts. Emerging data highlighting the therapeutic potential of the NSP14 ExoN activity has motivated the development of high-throughput nuclease assays to facilitate drug discovery. One example takes advantage of the small molecule intercalator RiboGreen that binds to dsRNA, the substrate for NSP14. Digestion of dsRNA by NSP14 ExoN activity dissociates the RiboGreen from the dsRNA and leads to a loss of fluorescence signal (Figure 1A) [37]. The RiboGreen format was used to screen 5000 small molecules against the NSP14-NSP10 complex to identify ExoN inhibitors. While the assay performance was sufficiently robust (Z-factor = 0.72), the readout suffers from two major limitations. Firstly, RiboGreen fluorescence decreases over time, which could lead to false negative results if plates are not read in a timely manner. Secondly, this assay is prone to high rates of false positives and false negatives, either due to compounds that displace the RiboGreen from dsRNA independent of nuclease activity or through intrinsic auto-fluorescence of small molecule compounds. These assay artifacts made it difficult to distinguish true inhibition hits, motivating the development of a FRET-based assay (Figure 1B) [37]. Utilizing a dsRNA sequence with a low melting temperature and featuring a fluorescent emitter on one strand and a fluorescent quencher on the other, ExoN activity releases the emitter generating a fluorescent signal. The FRET format is straightforward and accessible but is still prone to false positive and false negative results due to optical interference (auto-fluorescence or quenching) of small molecule library compounds. To overcome optical interference, a label-free and high-throughput assay has been described using a combination of self-assembled monolayers and matrix-assisted laser desorption ionization (MALDI) mass spectrometry, a technique termed SAMDI (Figure 1C). The SAMDI-MS assay offers several benefits, including a robust format (Z-factor = 0.85), significant signal-to-background ratio (>200), and the ability to distinguish multiple reaction products. The SAMDI-MS assay was used to screen 10,000 small molecules against the NSP14-NSP10 complex focused on the ExoN activity [38]. ExoN inhibitor selectivity was further evaluated against another RNA nuclease and in a thiazole orange assay to rule out RNA intercalators, revealing single-digit micromolar inhibitors that remain under investigation. This particular assay format is interesting given its multiplexing capability [39]. In this format, one could envisage monitoring the polymerase activity of RdRp and the proofreading activity of NSP14 ExoN to monitor the polymerase and proofreading activities of the complex simultaneously, offering a powerful assay to discover antiviral compounds against SARS-CoV-2 and other coronaviruses.

## 4. Known Small Molecules That Interfere with the Proofreading Activity of NSP14

### 4.1. Nucleosides Causing Delayed Chain Termination or Error Catastrophe

Until now, the main efforts to develop nucleoside analogs against SARS-CoV-2 have focused on repurposing molecules already known to be effective against other RNA viruses. However, obligate chain-terminating antiviral nucleosides are generally not effective against SARS-CoV-2 because coronaviruses carry the NSP14-encoded ExoN proofreading activity that removes incorporated nucleotide analogs from the 3′-end of nascent RNA (for a review, see [40]). Sofosbuvir, an approved hepatitis C virus drug that acts as an immediate chain terminator, was predicted in silico to be recognized in its nucleoside triphosphate form (SOF-TP) by the polymerase complex of SARS-CoV-2 [41]. It was recently reported that SOF-TP, once incorporated into viral RNA in its monophosphate form, is efficiently excised by NSP14, thereby allowing RNA synthesis to resume [24]. However, SOF-TP was also shown experimentally to be a poor substrate for viral RNA incorporation [42], consistent with the lack of anti-CoV activity of sofosbuvir in cell culture [43,44]. For these reasons, it remains unclear whether the lack of sofosbuvir activity against SARS-CoV-2 is due to high discrimination by the viral polymerase or to excision by the viral proofreading mechanism. In contrast, the nucleoside analog remdesivir efficiently inhibited SARS-CoV-2 in cell culture and in animal models [45,46,47]. Remdesivir was the first FDA-approved drug for the treatment of patients with COVID-19 (Figure 2). Remdesivir might be able to circumvent the ExoN excision/repair activity of NSP14 because its incorporation does not immediately terminate elongation of the nascent RNA strand but instead stalls the polymerase complex after the addition of three more nucleotides [42,48,49,50,51].

The RNA proofreading activity of the NSP14 ExoN domain also protects coronaviruses from ribavirin. Ribavirin is a triazole nucleoside that was discovered 50 years ago [52]. The mechanism of antiviral activity of ribavirin is believed to involve lethal mutagenesis as a combined result of inosine monophosphate dehydrogenase inhibition resulting in nucleotide pool imbalance, together with insertion into the nascent viral RNA [53,54]. Consequently, many unrelated RNA viruses are sensitive to ribavirin [55,56,57]. However, ribavirin has no or limited antiviral activity against coronaviruses, including SARS [58,59,60]. The ExoN of CoV NSP14 was shown to excise ribavirin 5′-monophosphate after its incorporation into viral RNA, providing a potential explanation for its limited efficacy in vivo [21]. Consequently, both ribavirin and 5′-fluorouracil, another mutagenic agent, displayed increased antiviral potency against an ExoN-defective CoV mutant [61]. Favipiravir, in its ribosylated triphosphate form, is another mutagenic nucleoside that was also once considered a potential treatment for COVID-19. Favipiravir (T-705) is a purine-base analog with broad-spectrum antiviral activity that is approved in Japan for the treatment of severe influenza virus infection. The nucleoside triphosphate form of favipiravir is efficiently recognized by the polymerase complex of SARS-CoV-2 [62]. The mutagenic antiviral effect of favipiravir has also been reported in cell culture, but overall, its anti-CoV potency is weak [46,62,63]. More studies are needed to determine if the ExoN activity of NSP14 is responsible for the weak in vitro anti-CoV effect of favipiravir. In the clinic, treatment of COVID-19 patients with favipiravir resulted in a modest to marginal benefit in terms of viral clearance, clinical improvement, and mortality [64].

Molnupiravir is another broad-spectrum antiviral agent that acts as a mutagenic nucleoside. Unlike ribavirin, molnupiravir is active against coronaviruses. Molnupiravir (EIDD-2801, MK-4482) was first evaluated for the treatment of alphavirus infections, and it recently received emergency use authorization by the FDA for the treatment of COVID-19 (for a review, see [65]). Molnupiravir is the valine ester orally bioavailable prodrug of β-d-N4-Hydroxycytidine (NHC; EIDD-1931). NHC inhibits coronaviruses lacking ExoN proofreading activity similarly to their wild-type counterparts, suggesting an ability to evade the NSP14 ExoN activity [66]. In cell culture, NHC is >100-fold more potent than ribavirin and favipiravir against SARS-CoV-2 [63]. Molnupiravir improved pulmonary function and reduced virus titer and body weight loss in mice infected with SARS-CoV or MERS-CoV [67]. The increased G>A and C>U transition mutation frequency in viral genome caused by molnupiravir treatment suggested that the nucleoside was incorporated into viral RNA by bypassing NSP14 ExoN. This was confirmed in two independent mechanistic studies using purified viral polymerase complexes [68,69,70]. The authors showed that the active entity NHC 5′-triphosphate is an efficient substrate of SARS-CoV-2 RdRp complex without causing inhibition of RNA synthesis. Once incorporated into viral RNA, NHC-monophosphate supported the formation of both NHC:G and NHC:A base pairs with similar efficiencies. However, it should be noted that NHC also induced mutagenic effects on host genes, indicating a lack of selectivity between viral and host polymerases and suggesting potential safety liabilities [63].

### 4.2. Non-Nucleoside Inhibitors

A selection of non-nucleoside analog inhibitors of NSP14 ExoN has recently been described (Figure 3). In one example using the FRET-based assay described above, two compounds, patulin and aurintricarboxylic acid, were revealed as potential NSP14 ExoN inhibitors. The two compounds appeared to inhibit NSP14-NSP10 ExoN activity with selectivity over RNaseA and benzonase, and interestingly reduced viral proliferation in VERO E6 cell-based assays. When combined with the nucleoside analog remdesivir, patulin and aurintricarboxylic acid did not show synergistic effects. Since ExoN mutations have been shown to have higher sensitivity to remdesivir treatment [71], these data suggest that in cells the two compounds may operate through mechanisms other than targeting the ExoN activity. Another study utilized the FRET-based assay to test compounds following structural analysis [72]. Several compounds were identified with IC_50_ values around 20 µM in biochemical assays that also exhibited synergistic activity with remdesivir in the viral assay [72]. Similar synergistic results were recently obtained when combining nucleoside analogs with HCV NS5A inhibitors also blocking SARS-CoV-2 NSP14 [73]. It is important to note that the biological mechanism underlying the synergistic effect remains to be defined. Future work combining pharmacologic and genetic approaches will shed light on whether these compounds and others target the NSP14 ExoN activity and proofreading mechanism as well as their therapeutic potential as antiviral agents.

The available structural data have motivated several researchers to focus on in silico screening approaches to reveal potential inhibitors. In one study, a homology model of the NSP14-NSP10 complex bound to RNA was used to virtually screen over 5000 compounds, including FDA-approved drugs, natural products, and an antiviral library of approximately 300 compounds. The study revealed a number of potential binders, including the RNA bases guanosine and inosine, the chemotherapeutic carfilzomib, and the antiviral compound ritonavir. The predicted binding pocket of ritonavir overlaps with the RNA binding pocket, suggesting that ritonavir may inhibit the ExoN activity [74]. The study postulates several molecules with therapeutic potential, but future biochemical and cellular assays are required to properly assess the activity of the compounds. The structural data of NSP14 interacting partners, including NSP10 and other members of the replication transcription complex, open avenues for in silico screening of compounds that may disrupt critical protein-protein interactions. A novel comprehensive in silico screening approach, VirtualFlow [75], was recently used in a discovery program to identify small molecules that may bind to various sites of SARS-CoV-2 proteins, including NSP14 [76]. The study relied on the available structural data for 15 viral proteins and two human proteins. An advantage of this particular approach was its ability to assess multiple potential binding sites on each protein, allowing the identification of non-competitive binders that could be used in combination for improved efficacy or to overcome future variants. For the NSP14 ExoN site, betamethasone was identified as a potential binder, suggesting it may have therapeutic value beyond the anti-inflammatory effects in the host commonly seen with corticosteroids, although this theory remains to be validated experimentally. While virtual screens can potentially assess billions of compounds in weeks to months, there are several challenges to consider. First, the success of the screen depends on the available structural data, which may differ in the biochemical and cellular context. Second, the binding affinities are calculated based on docking scores and the Gibbs’ equation and therefore should be considered approximations rather than actual dissociation constants. Third, biochemical and cell-based assays are still required to validate the predicted hits and demonstrate proof of target engagement.

## 5. Conclusions: Is There a Case for NSP14 Proofreading Inhibitors?

The recent approval of the influenza polymerase cap-snatching inhibitor baloxavir marboxil (Xofluza) provides the first proof of principle that viral nucleases are viable targets for antiviral therapies. In contrast, there are very few reports of small molecules inhibiting the ExoN activity of the CoV replication complex. Biochemical ExoN assays to support compound screening and profiling have been described, but the biggest limitation seems to be the ability to monitor the effect of ExoN inhibition in infected cells. As recently described, it is likely that ExoN inhibitors might not have a direct antiviral effect on their own, but would need to be combined with other modalities such as nucleoside analogs [72]. The ability to generate resistance mutations with this new class of anti-CoV agents would also provide further evidence of target engagement in a cellular context. Finally, we cannot neglect the theoretical concern of increased mutations leading to the emergence of new variants as a result of inhibiting the proofreading function of the SARS-CoV-2 replication complex. Taken together, pursuing NSP14 proofreading inhibitors remains an encouraging therapeutic avenue. New studies to further validate the target and characterize the proofreading activity in a cellular context and in animals will be needed.

## Figures and Tables

**Figure 1 molecules-27-02918-f001:**
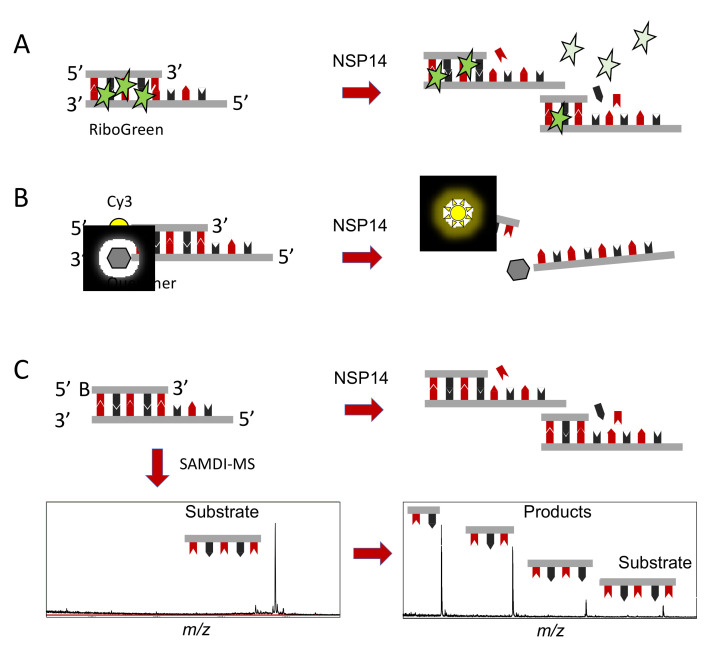
High-throughput biochemical activity assays for NSP14 exonuclease activity. (**A**). Double-stranded RNA is incubated with the non-specific intercalator RiboGreen. Upon nuclease activity, the RiboGreen is released, corresponding to a loss of signal. (**B**). Traditional FRET assay using dsRNA substrates featuring a fluorescent emitter, such as Cy3, and a fluorescent quencher. Upon NSP14 nuclease activity, the two strands dissociate, releasing the fluorescent emitter generating a fluorescent signal. (**C**). A dsRNA substrate featuring a biotin on the 5′ end is digested by NSP14 exonuclease activity. The biotinylated substrates and products are immobilized onto NeutrAvidin, presenting self-assembled monolayers in a high-density biochip array format that are efficient substrates for MALDI-ToF-MS, a technique termed SAMDI.

**Figure 2 molecules-27-02918-f002:**
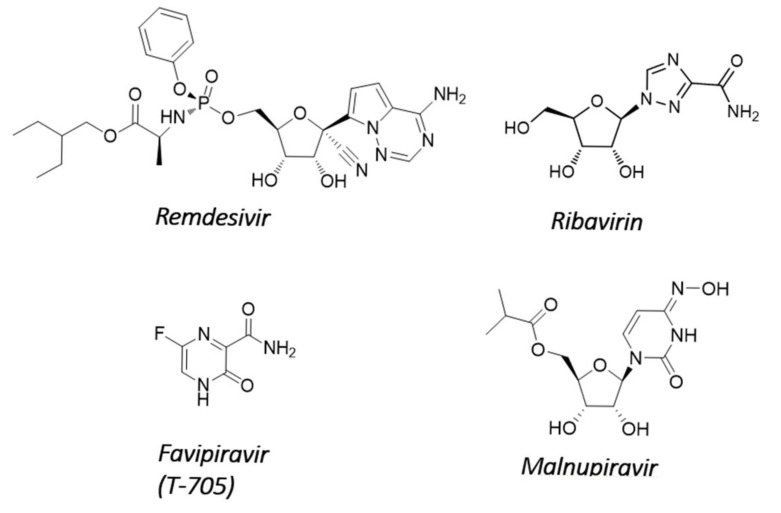
Representative nucleoside analogs interfering with RNA proofreading.

**Figure 3 molecules-27-02918-f003:**
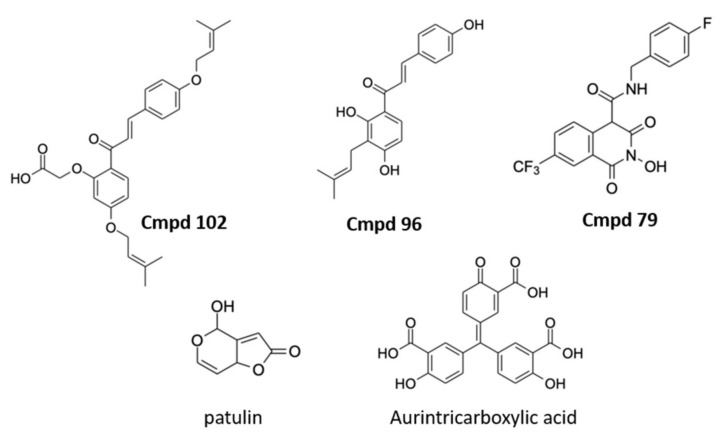
Representative non-nucleoside NSP14 inhibitors. Compounds **102** (IC_50_: 19.4 μM), **96** (IC_50_: 17.4 μM), and **79** (IC_50_: 22.0 μM) described in [72], and patulin (IC_50_: 1.8 μM) and aurintricarboxylic acid (IC_50_: 10.3 μM) described in [37].

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
