# Peer review of "Opportunities and Challenges in Targeting the Proofreading Activity of SARS-CoV-2 Polymerase Complex"

_molecules, 2022, doi:10.3390/molecules27092918_

Round 1
Reviewer 1 Report
Overall the article by Deval and Gurard-Levin is an informative review on the nsp14 ExoN activity that is nicely written and clearly deserves to be published in Molecules. However, some small concerns should be answered prior to publication.
1)The RNA genome of SARS-CoV-2 is divided into multiple open-reading frames, among which ORF1ab contains the nonstructural proteins (NSPs) 7 to 16 required for RNA replication and transcription.
True. But it sounds little bit misleading. ORF1ab encodes for nsp1-16 of which 7-16 are required for RNA replication with the exception of nsp11.
2) NSP12 also encodes a second enzymatic site…
True. Perhaps it would be worth mentioning that the NiRAN domain is also responsible for UMPylation of the nsp9?
3) While the G7-MTase role in RNA capping and its relevance as a therapeutic target have been described elsewhere [7],[8],[9]
Real nanomolar inhibitors were described by Otava et al. and also by the Vedadi lab.
4) This process is further supported by recent modeling pro-posing that nascent double stranded RNA exiting the polymerase passes over the NSP14 ExoN and NSP15 endonuclease sites before being unwound [21].
I believe there is also a cryo-EM structure of this complex available. Check the recent review by Nencka et al.
5) chain-terminating antiviral nucleosides are generally not effective against SARS-CoV-2
Delayed chain terminators (Remdesivir) seems pretty effective. Just change to: Obligate chain-terminating...
6) Sofosbuvir, an approved hepatitis C virus drug that acts as an immediate chain terminator...
That is probably fine, however, I believe obligate chain-terminator is more appropriate here.
7) It is hypothesized t hat remdesivir circumvents the ExoN excision/repair activity of NSP14 because its incorporation does not immediately terminate elongation of the nascent RNA strand but instead stalls the polymerase complex after the addition of three more nucleotides
Not hypothesized. That is a well-established fact.
8) 4. Known small molecules that interfere with the proofreading activity of NSP14
The hepatitis C NS5A inhibitors including velpatasvir and elbasvir, which were later shown to act on nsp14 should be discussed here.
9) The recent approval of the influenza polymerase cap snatching inhibitor baloxavir marboxil (Xofluza) provides proof of principle that viral nucleases are viable targets for antiviral therapies.
This sounds more like a proof that the capping machinery, which in most viruses including CoVs does not have any nuclease, is a viable target.
10) Several new crystallographic studies of nsp14 have just appeared – probably after the manuscript was written. However, perhaps their findings could strengthen this review article?
Author Response
Reviewer 1:
Overall the article by Deval and Gurard-Levin is an informative review on the nsp14 ExoN activity that is nicely written and clearly deserves to be published in Molecules. However, some small concerns should be answered prior to publication.
1)The RNA genome of SARS-CoV-2 is divided into multiple open-reading frames, among which ORF1ab contains the nonstructural proteins (NSPs) 7 to 16 required for RNA replication and transcription.
True. But it sounds little bit misleading. ORF1ab encodes for nsp1-16 of which 7-16 are required for RNA replication with the exception of nsp11.
RESPONSE: thank you for catching this imprecision. The appropriate change has been made.
2) NSP12 also encodes a second enzymatic site…
True. Perhaps it would be worth mentioning that the NiRAN domain is also responsible for UMPylation of the nsp9?
RESPONSE: this information has been added.
3) While the G7-MTase role in RNA capping and its relevance as a therapeutic target have been described elsewhere [7],[8],[9]
Real nanomolar inhibitors were described by Otava et al. and also by the Vedadi lab.
RESPONSE: the 2 references have been added.
4) This process is further supported by recent modeling pro-posing that nascent double stranded RNA exiting the polymerase passes over the NSP14 ExoN and NSP15 endonuclease sites before being unwound [21].
I believe there is also a cryo-EM structure of this complex available. Check the recent review by Nencka et al.
RESPONSE: The references to cryo-EM structures have now been added.
5) chain-terminating antiviral nucleosides are generally not effective against SARS-CoV-2
Delayed chain terminators (Remdesivir) seems pretty effective. Just change to: Obligate chain-terminating...
RESPONSE: Change made.
6) Sofosbuvir, an approved hepatitis C virus drug that acts as an immediate chain terminator...
That is probably fine, however, I believe obligate chain-terminator is more appropriate here.
RESPONSE: we would prefer to leave the term immediate instead of obligate because sofosbuvir has a 3’-OH group.
7) It is hypothesized that remdesivir circumvents the ExoN excision/repair activity of NSP14 because its incorporation does not immediately terminate elongation of the nascent RNA strand but instead stalls the polymerase complex after the addition of three more nucleotides
Not hypothesized. That is a well-established fact.
RESPONSE: We would prefer to be a bit more nuanced. Although there are several reports of the non-immediate chain termination effect of remdesivir, to our knowledge there is no clear data showing that RNA containing remdesivir is able to circumvent the ExoN activity of NSP14. Therefore we changed the sentence to: “Remdesivir might be able to circumvent…”
8) 4. Known small molecules that interfere with the proofreading activity of NSP14
The hepatitis C NS5A inhibitors including velpatasvir and elbasvir, which were later shown to act on nsp14 should be discussed here.
RESPONSE: A sentence was added to mention the hepatitis C NS5A inhibitors inhibiting NSP14.
9) The recent approval of the influenza polymerase cap snatching inhibitor baloxavir marboxil (Xofluza) provides proof of principle that viral nucleases are viable targets for antiviral therapies.
This sounds more like a proof that the capping machinery, which in most viruses including CoVs does not have any nuclease, is a viable target.
RESPONSE: we agree that CoVs don’t have a cap snatching activity, however we think it is worth mentioning that the influenza endonuclease inhibitor example provides some level of validation for targeting any viral nuclease (endo- or exo-) function associated with the polymerase complex.
10) Several new crystallographic studies of nsp14 have just appeared – probably after the manuscript was written. However, perhaps their findings could strengthen this review article?
RESPONSE: Thank you for pointing out additional structural studies. We have included recent publications featured in JBC and a new article in PNAS (Moeller et al., PNAS 2022).

Reviewer 2 Report
This manuscript reviews on the "Opportunities and Challenges in Targeting the Proofreading Activity of SARS-CoV-2 Polymerase Complex", a complex that can serve as a target for future development of anti-Covid agents. The role of nsp14 protein is described and some examples of derivatives acting on it are also provided. The manuscript is well written and includes necessary information about the role and the therapeutic potential of nsp14. Thus, it could be considered for publication, after minor revision.
Minor issues to be addressed:
-The crystal structure of SARS-CoV-2 ExoN with nsp10 has been recently determined (Moeller et al., PNAS 2022, 119(9),e2106379119). This information can serve as a platform for future development of drugs against Covid19 infection or strategies to attenuate the viral virulence. Thus, this data should be included in this review.
-Addition of a figure with the structures of the nucleoside derivatives discussed in section 4.1 is highly recommended.
-The authors should provide some more information, for example IC50s, concerning the activity of the inhibitors discussed in sections 4.1 and 4.2.
-Page 4, line 157: ‘Ribavirin is a purine analog’ should be replaced by ‘Ribavirin is a triazole nucleoside’.
-Page 4, line 167: ‘Favipiravir is another mutagenic nucleoside’ should be replaced by ‘Favipiravir, in its ribosylated triphosphate form, is another mutagenic nucleoside’.
Author Response
Reviewer 2:
This manuscript reviews on the "Opportunities and Challenges in Targeting the Proofreading Activity of SARS-CoV-2 Polymerase Complex", a complex that can serve as a target for future development of anti-Covid agents. The role of nsp14 protein is described and some examples of derivatives acting on it are also provided. The manuscript is well written and includes necessary information about the role and the therapeutic potential of nsp14. Thus, it could be considered for publication, after minor revision.
Minor issues to be addressed:
-The crystal structure of SARS-CoV-2 ExoN with nsp10 has been recently determined (Moeller et al., PNAS 2022, 119(9),e2106379119). This information can serve as a platform for future development of drugs against Covid19 infection or strategies to attenuate the viral virulence. Thus, this data should be included in this review.
RESPONSE: Thank you for pointing out this new important article. We have now referenced it in our review manuscript.
-Addition of a figure with the structures of the nucleoside derivatives discussed in section 4.1 is highly recommended.
RESPONSE: a new figure 2 has been added to show the structures of relevant nucleoside derivatives.
-The authors should provide some more information, for example IC50s, concerning the activity of the inhibitors discussed in sections 4.1 and 4.2.
RESPONSE: We have included the IC50 values in the Figure Legend for the in vitro non-nucleoside inhibitors reported. The nucleoside-based inhibitors often do not target the NSP14 enzyme directly, but rather the polymerase enzyme, and therefore are not reported as this review focuses on NSP14.
-Page 4, line 157: ‘Ribavirin is a purine analog’ should be replaced by ‘Ribavirin is a triazole nucleoside’.
RESPONSE: change made.
-Page 4, line 167: ‘Favipiravir is another mutagenic nucleoside’ should be replaced by ‘Favipiravir, in its ribosylated triphosphate form, is another mutagenic nucleoside’.
RESPONSE: change made.
